# Promising Approaches Based on Bioimaging Reporters for Direct Rapid Detection of *Mycobacterium tuberculosis*

**DOI:** 10.3390/biomedicines13112609

**Published:** 2025-10-24

**Authors:** Oganes A. Ambartsumyan, Olesya A. Skuredina, Platon I. Eliseev, Tatiana E. Tiulkova, Anastasia G. Samoilova, Irina A. Vasilieva

**Affiliations:** 1National Medical Research Center of Phthisiopulmonology and Infectious Diseases of the Health Ministry of the Russian Federation, Moscow 127473, Russia; inori1626@gmail.com (O.A.S.); pediatrics@yandex.ru (P.I.E.); tulkova2006@rambler.ru (T.E.T.); a.samoilova.nmrc@mail.ru (A.G.S.); vasil39@list.ru (I.A.V.); 2Osipyan Institute of Solid State Physics RAS, Chernogolovka 142432, Russia

**Keywords:** *M. tuberculosis*, tuberculosis diagnosis, tuberculosis diagnostics, molecular reporters, molecular probes, fluorogenic probes, no-wash imaging, point-of-care

## Abstract

Tuberculosis remains a serious global public health challenge and requires the development of rapid, sensitive, and specific diagnostic tools for effective treatment and disease control. Bioimaging reporters are promising diagnostic tools that exploit the unique biochemical properties of *Mycobacterium tuberculosis* for real-time detection of viable cells from clinical samples. Moreover, these methods offer significant advantages over the conventional methods currently used in practice, including reduced assay time, increased specificity, and the ability to discriminate viable cells from dead cells. In this review, we highlight reporters of a different nature that the enable direct detection of *Mycobacterium tuberculosis*, eliminating complex sample preparation. Such reporters could serve as powerful tools in fluorescence microscopy, provide alternative strategies for automated culture-based diagnostic systems, and offer new approaches for developing point-of-care methods and diagnostic devices suitable for clinical practice.

## 1. Introduction

Despite progress in tuberculosis (TB) control, challenges remain in the timely diagnosis and detection of antibiotic resistance worldwide [1]. The main methods for detecting *Mycobacterium tuberculosis* (*Mtb*) are sputum smear microscopy, molecular methods, and culture-based methods [2,3,4,5].

Microscopy remains an important tool for the diagnosis of TB, especially in resource-limited settings, as it is a rapid, relatively simple, inexpensive, and sufficiently specific method [6,7]. However, it has several limitations, including low sensitivity, an inability to differentiate between viable and non-viable cells or to distinguish between *Mtb* and nontuberculous mycobacteria (NTM), and a lack of information on drug susceptibility [8,9,10]. Therefore, microscopy results require additional testing via culture-based or molecular methods to ensure accurate diagnosis and guide appropriate treatment.

The use of molecular diagnostic methods, such as GeneXpert (Cepheid, Sunnyvale, CA, USA), has significant advantages, including rapid identification directly from clinical samples, high sensitivity and specificity, and the ability to identify mutations associated with drug resistance in key target genes [3,11,12].

Additionally, molecular tests cannot distinguish viable from nonviable organisms, as they detect genetic material regardless of viability, which may lead to challenges in interpreting results, especially in patients who have recently completed treatment [13,14].

The use of molecular methods also requires qualified personnel, specialized equipment, and financial resources, potentially limiting their availability in low-resource settings. Furthermore, the clinical significance of some detected mutations may not always be clear, complicating treatment decisions. Because of these factors, molecular diagnostics are often used in conjunction with traditional phenotypic drug susceptibility testing (DST) to provide a more comprehensive understanding of resistance patterns and to guide effective patient management [14,15].

Culture-based methods using solid media remain the cornerstone of tuberculosis diagnostics [16,17,18]. These methods allow for the detection of viable *Mtb* cells and enable drug susceptibility testing with high sensitivity and affordability. However, their main limitation is the long turnaround time required for bacterial growth, as well as their limited ability to distinguish *Mtb* from NTM [3,19].

Compared with conventional solid media cultivation, automated liquid culture systems reduce the turnaround time for *Mtb* detection in DSTs [20,21]. An example of such a system is the BACTEC MGIT 960 (Becton Dickinson, Sparks, MD, USA), an automated culture-based diagnostic tool that detects bacterial growth by measuring oxygen consumption [22,23] in the test tube; oxygen depletion releases quenching of a fluorochrome embedded in the tube, resulting in fluorescence proportional to mycobacterial growth.

Although the use of liquid culture systems such as BACTEC MGIT 960 shortens the analysis time relative to that of solid media, they still have limitations in terms of turnaround time and cannot differentiate *Mtb* from other mycobacteria; therefore, the continued search for novel, rapid, and specific diagnostic approaches for *Mtb* detection remains crucial.

In recent years, there has been a significant increase in research focused on developing new approaches for the rapid detection of bacterial pathogens on the basis of various reporter systems for bioimaging [24,25,26,27,28,29,30]. These approaches primarily involve the use of fluorescent or fluorogenic reporters, as well as fluorescent proteins.

Most fluorescent reporters are small-molecule probes [31,32,33,34,35] that can be classified according to their targets as enzyme-activated, membrane- or cell wall component-targeting, or metabolic-targeting probes [36,37]. The strategy based on the use of fluorescent proteins involves engineering bacteriophages to carry and express genes encoding fluorescent proteins. Upon infecting the target bacteria, these phages introduce the fluorescent protein-encoding gene into the bacterial host, leading to the expression of fluorescent proteins inside the bacterial cells [38].

Commonly used fluorescent reporter systems for *Mtb* include fluorogenic and chemiluminescent probes, as well as genetically engineered reporter phages (fluorescent bacteriophages, fluorophages) [39,40,41,42,43].

For possible clinical diagnostic applications, significant attention has been given to reporters with a “turn-on” mechanism due to their minimal background signal in the unreacted state and their activation only upon interaction with target molecules. This enables real-time detection of living cells without the need for washing steps after labeling [44,45,46].

The “turn-on” mechanism can be achieved through different strategies, including the use of probes containing fluorogenic dyes that become fluorescent in response to local environmental changes such as solvent polarity, viscosity, or pH [47,48,49,50,51,52]; the application of fluorophore‒quencher pairs that switch from quenched to fluorescent states upon separation [53,54,55,56,57,58,59,60,61]; or introduction fluorescent protein reporter genes [62] into bacterial cell, that is primarily achieved through the use of genetically engineered bacteriophages (reporter phages) [63,64].

In this review, we examine promising approaches using various types of wash-free bioimaging reporters for the rapid detection of *Mtb*.

## 2. Fluorophore‒Quencher Pair Probes

Currently, a significant portion of fluorophore‒quencher pair probes for bacterial detection are enzyme-activated probes that use pathogen-specific enzymes as detection targets [65,66,67,68].

These probes typically consist of three key components: a light-emitting group, a linker, and an enzyme recognition unit, which usually acts as a quencher. The fluorophore generates a detectable signal upon activation, whereas the linker connects the fluorophore to the recognition unit. The recognition unit is a specific molecular substrate or chemical group designed for selective recognition and cleavage or transformation by the target enzyme, which relieves fluorescence quenching and results in probe activation [60,69].

### 2.1. FLASH Hip1 Probe

The use of luminescence, particularly chemiluminescence and bioluminescence, as a detection method is widely established in microbiology, including in the development of novel diagnostic approaches [70,71,72,73]. These techniques rely on the ability of specific substances and biological systems to emit light as a result of chemical or biochemical reactions, enabling sensitive and effective identification and analysis of microorganisms and their components [74].

One of the key immunomodulatory virulence factors of *Mtb* is the serine protease Hip1 (hydrolase important for pathogenesis 1), which reduces macrophage production of proinflammatory cytokines, inhibits dendritic cell maturation, and impairs antigen presentation and T-cell responses [75,76,77,78,79].

Babin et al. chose Hip1 as a target to develop a chemiluminescent protease probe, which consists of a tetrapeptide Hip1 substrate linked to a phenoxy-dioxetane luminophore via a self-eliminating linker [80] (Figure 1).

Upon enzymatic cleavage, the aniline linker undergoes spontaneous elimination, resulting in the release of the luminophore and subsequent light emission. The experiment was conducted using two strains of *Mtb*, the laboratory strain H37Rv and the attenuated strain mc^2^6020, both of which were cultured in Middlebrook 7H9 medium. The limits of detection for the H37Rv and mc^2^6020 strains were 23,000 and 4000 cells, respectively. Additionally, an experiment assessing the detection of *Mtb* in sputum demonstrated a limit of detection of 15,000 cells for the mc^2^6020 strain. To evaluate the specificity of the probe, luminescence signals were measured using equal amounts of *Mtb* and NTM. The results revealed that the luminescence intensity for *Mtb* was greater than that for NTM, indicating that the probe is highly specific to *M. tuberculosis*. The potential of the assay for the DST was also demonstrated. An experiment was conducted to compare the luminescence signals of a rifampicin-sensitive strain H37Rv and a rifampicin-resistant strain (RpoB H526D mutant) over several days. The results revealed that the luminescence signal from the rifampicin-resistant strain increased over time and was significantly greater than that of the rifampicin-sensitive strain treated with rifampicin.

### 2.2. CDG-DNB3 Probe

Another enzyme-activated, no-wash probe for the rapid detection of *Mtb* is CDG-DNB3, developed by Yunfeng Cheng et al. [81].

This probe employs a dual-targeting strategy by simultaneously targeting two key bacterial enzymes: β-lactamase (BlaC) and decaprenylphosphoryl-β-d-ribose-2′-epimerase (DprE1), an essential enzyme involved in cell wall biosynthesis. β-Lactamases are enzymes produced by certain bacteria that provide resistance to β-lactam antibiotics by hydrolyzing the β-lactam ring [82,83].

Using BlaC as a target is a common strategy for developing different types of probes [84,85,86,87]. These probes incorporate substrates or chemical groups that are specifically recognized and cleaved by β-lactamases, resulting in fluorescence activation. Probes targeting the BlaC enzyme typically produce an ‘off-on’ fluorescence signal upon interaction with β-lactamase, enabling sensitive and selective real-time detection of bacteria [88,89,90,91,92,93]. This principle was used for developing the CDG-DNB3 probe.

Upon entry into *Mtb* cells, BlaC hydrolyzes the β-lactam ring in CDG-DNB3, releasing the fluorescent reporter and turning the signal ‘on.’ Meanwhile, DprE1 reduces a nitro group to form a nitroso derivative, creates a stable semimercaptal complex, and traps the fluorophore inside the cell. CDG-DNB3 enables differentiation of *Mtb* from 43 other mycobacterial species and distinguishes live mycobacteria from dead. The probe can detect the pathogen in sputum samples within one hour and, in some cases, outperforms traditional Auramine O staining.

This integrated approach offers a rapid, sensitive and specific method for live *Mtb* detection and quantification, with promising applications in tuberculosis diagnosis and treatment monitoring. Notably, CDG-DNB3 demonstrated higher sensitivity than auramine O in sputum samples, detecting *Mtb* in GeneXpert-positive but auramine O-negative cases.

### 2.3. NFC-Probe

One of *Mtb*’s enzymatic defenses against external factors, such as the host immune response, are nitroreductases [94,95] which can be used as targets for developing di-agnostic probes. Mu et al. developed NFC-probe activated by the *Mycobacterium tuberculosis*-specific Rv2466c nitroreductase [96]. This process occurs with the participation of mycothiol, a unique low-molecular-weight thiol present in actinomycetes, including mycobacteria. Mycothiol binds to the nitroreductase, inducing con-formational changes and forming an active catalytic center. Within this active site, multistep reduction of the probe’s nitro group to an amino group occurs, with mycothiol serving as both electron and proton donor. Following nitro group reduction, fluorescence intensity increases by 220-fold. The signal can be detected using fluorescence microscopy or fluorometer, with fluorescence intensity directly correlating with the number of viable and metabolically active bacteria in the sample.

Detection limits in pure cultures of clinical isolates *Mtb* range from 1.5 × 10^4^ cells/mL for extensively drug-resistant tuberculosis (XDR-TB) strains to 1.2 × 10^6^ cells/mL for multidrug-resistant tuberculosis (MDR-TB) strains. Bacterial detection time is 5 days, including 24 h sample incubation with the probe, which is significantly faster than traditional culture-based methods. Additionally, the NFC probe enables determination of minimum inhibitory concentrations (MICs) for anti-tuberculosis drugs, facilitating rapid DST. Subsequently, advanced NFC probe variants were developed—including the trehalose-conjugated NFC-Tre-5, which exhibits enhanced specificity [97] and 17a-Tre probe [98], which delivers a 20-fold signal amplification and reduced response time.

### 2.4. Cy3-NO_2_-Tre

Another nitroreductase-targeting approach involves a fluorescent probe developed by Hong et al. developed an Rv3368c-targeted probe, Cy3-NO_2_-tre.

It consists of a cyanine fluorophore, a nitroaromatic group that serves as a fluorescence quencher, and trehalose, which allows specific probe uptake by viable mycobacterial cells [99].

Rv3368c is an oxidoreductase present in various mycobacterial species. It is upregulated under oxygen-starved conditions and during nonreplicating states of mycobacteria, suggesting a potential role in persistence [100].

The fluorescence “turn-on” mechanism in this probe is based on the enzymatic reduction of the nitro group by the nitroreductase Rv3368c to an amine, that eliminates quenching and activates the fluorescence.

The ability of the probe to target mycobacteria was evaluated in clinical samples by adding it to decontaminated sputum and incubating at 37 °C for 15 min, 1 h, 2 h, and 24 h, followed by direct evaluation without washing steps. A fluorescence signal was detected as early as 15 min, with an intensity at 2 h comparable to that at 24 h, and the limit of detection of the assay was 4.3 × 10^2^ CFU (colony-forming units).

The specificity test of Cy3-NO_2_-tre revealed that, compared with other bacterial species, the probe selectively and strongly labels *M. smegmatis*. In the experiments, incubation of *M. smegmatis* with 1.0 μM Cy3-NO_2_-tre at 37 °C for 1 h resulted in specific fluorescence labeling, whereas other bacteria, such as *E. coli*, *E. faecalis*, *S. aureus*, *B. subtilis*, *L. monocytogenes*, and *S. pneumoniae*, showed negligible staining.

Given the demonstrated sensitivity and specificity in this study and the lack of additional washing steps, Cy3-NO_2_-tre and the principle on which it is based are promising for the development and application of similar solutions for the rapid detection of *Mtb.*

### 2.5. N14G and N14G-Fe

Iron is an essential component of the metabolism and growth of *Mtb* [101,102,103,104]. To scavenge Fe^3+^ ions from the external environment, mycobacteria synthesize and use two types of siderophores: lipid-bound mycobactin and soluble carboxymycobactin [105]. After Fe^3+^ binds to the extracellular space, iron-chelated mycobactin is actively imported into mycobacterial cells via iron-regulated transporter AB (IrtAB) [106,107,108].

This process is coupled with the intracellular reduction of Fe^3+^ to Fe^2+^ by the siderophore interaction domain (SID), resulting in the release of Fe^2+^ and free mycobactin. The active transport mechanism through IrtAB, combined with the subsequent iron reduction and release process, represents a highly specific pathway that can be exploited as a target for fluorescent diagnostic probes [109,110,111].

Dianmo Ni et al. developed a novel mycobactin-fluorophore (rhodamine)-conjugated probe (MbTFCp), designated N14G, and its Fe^3+^-chelated form, N14G-Fe [112].

The latter is distinguished by Fe^3+^ ions capable of quenching fluorescence in the external environment but activated fluorescence within mycobacteria following the reduction and release of Fe^2+^ ions (Figure 2).

The ability of the probes to label *Mtb* cells was tested on the laboratory strain H37Rv. Both probes demonstrated the ability to label H37Rv cells within 10 min, with a limit of detection of 34 CFU for N14G-Fe.

Furthermore, the ability of the probes to detect *Mtb* in clinical samples was demonstrated. N14G and N14G-Fe probes were added at a concentration of 0.1 µM to decontaminate the sputum samples, which were then incubated for 10 min and subsequently analyzed for fluorescence signals. Both probes produced comparable fluorescence levels, achieving a limit of detection of 34 CFU for the H37Rv strain. Notably, the use of the N14G-Fe probe does not require a washing step because of its low background fluorescence, simplifying the detection protocol.

Specificity analysis of N14G and N14G-Fe using *M. smegmatis*, *B. subtilis*, *E. coli*, *S. aureus*, and *L. monocytogenes* demonstrated that the fluorescence intensity of N14G-labeled *M. smegmatis* was greater than that of the other bacteria. Similarly, iron-chelated N14G-Fe produced a stronger signal in *M. smegmatis* than in the other species. These results show that the IrtAB transport system is characteristic of mycobacteria, allowing it to be considered a potential target for the development of diagnostic probes.

## 3. Fluorogenic Probes

Trehalose serves several critical functions in Mycobacteria, particularly *Mtb*. As a structural component, trehalose is a core sugar in key mycobacterial glycolipids, such as trehalose dimycolate (TDM), also known as cord factor, and trehalose monomycolate (TMM), which are essential for the formation of the mycolic acid cell wall [113,114,115,116,117,118].

*Mtb* synthesizes trehalose de novo but also utilizes exogenous trehalose. The transport of trehalose molecules from the extracellular environment is mediated by the mycomembrane-associated protein PPE51, which facilitates passage through the outer layer of the mycobacterial mycomembrane [119].

Once in the periplasmic space, other transporters, such as the ABC transporter LpqY-SugABC [120,121,122,123], recycle trehalose back into the cytoplasm for further metabolism and incorporation into cell wall components via the action of the Ag85 enzyme complex [124,125,126]. Notably, the rate of trehalose uptake by mycobacteria is greater than that by other bacteria because of the significant role of trehalose in their metabolism [127]. This feature underlies the specificity of trehalose-based diagnostic probes [128,129].

Considering the key features of trehalose metabolism in mycobacteria, the trehalose uptake pathway has become an important target for developing trehalose probes—diagnostic tools based on chemically modified trehalose analogs conjugated to fluorogenic dyes (Figure 3) [130].

### 3.1. DMN-Tre

Mireille Kamariza and colleagues developed a fluorogenic trehalose probe, DMN-Tre, which consists of a 4-N,N-dimethylamino-1,8-naphthalimide solvatochromic dye conjugated to trehalose and designed for the rapid detection of *Mtb* [131]. The probe functions by metabolically incorporating DMN-Tre into the mycomembrane, where its fluorescence in the hydrophobic environment increases by more than 700-fold. The probe exhibits a turn-on fluorescence mechanism, becoming highly fluorescent upon partitioning into the hydrophobic membrane environment while remaining weakly fluorescent in aqueous solutions.

The key advantages of DMN-Tre include a no-wash protocol, rapid labeling capability, and the ability to distinguish viable mycobacteria from non-viable mycobacteria and to determine antibiotic sensitivity. The limit of detection was approximately 10,000 CFU/mL in pure cultures of *M. smegmatis*, which is comparable to auramine staining. Notably, the fluorescent signal could be observed as early as 5 min after incubation of *M. smegmatis* with the probe. In sputum samples, a fluorescent signal from *Mtb* was detected within 1 h of incubation with DMN-Tre, which was similar to auramine staining but required simpler processing, demonstrating its potential for clinical application.

In specificity studies, the probe showed no labeling when incubated with Gram-negative *E. coli* or Gram-positive *S. aureus*, *L. monocytogenes*, or *B. subtilis*. Remarkably, the probe’s fluorescence is inhibited by antitubercular drugs, making it a promising tool for drug susceptibility testing.

### 3.2. HC-3-Tre

The subsequent development by Mireille Kamariza, Carolyn R. Bertozzi et al. involved solvatochromic trehalose probes, specifically 3HC-2-Tre and 3HC-3-Tre, which are conjugates of 3-hydroxychromone (3HC) dyes and trehalose [128]. These probes exploit the metabolic incorporation of trehalose into the mycobacterial cell wall by Ag85 enzymes.

3HC dyes are solvatochromic compounds that exhibit shifts in fluorescence intensity in response to changes in environmental polarity. When conjugated to trehalose, these probes are specifically processed by the Ag85 enzyme complex, which catalyzes the mycolylation of trehalose at the 6-position to form trehalose monomycolate (TMM). This enzymatic modification enables the incorporation of the probe into the hydrophobic mycomembrane environment, where the resulting change in polarity leads to fluorescence turn-on.

Compared with the previously reported DMN-Tre probe, 3HC-3-Tre demonstrated a 10-fold increase in fluorescence intensity and enabled the detection of *Mtb* within 10 min in pure culture. Additionally, 3HC-3-Tre shows minimal background fluorescence and eliminates the need for wash steps.

In contrast, while 3HC-2-Tre labels mycobacteria with approximately 100-fold higher fluorescence intensity than DMN-Tre does, it has limited specificity, as it also binds nonspecifically to other bacterial species. Furthermore, 3HC-2-Tre exhibited rapid washout kinetics, reflecting trehalose-independent transient binding mechanisms.

3HC-3-Tre demonstrates high specificity for mycobacteria and closely related Actinobacteria that utilize trehalose in their cell envelope biosynthesis. The probe efficiently labels *M. smegmatis* and *C. glutamicum*, both of which incorporate trehalose into their cell wall structures. In contrast, non-Actinobacteria species, including *B. subtilis*, *E. coli*, and *S. aureus*, show minimal fluorescence, likely representing trehalose-independent interactions that can be readily distinguished from specific labeling. The probes are highly sensitive and detect *Mtb* at low concentrations (e.g., 0.05 mM for 3HC-3-Tre). Rapid, wash-free detection and high brightness make 3HC-3-Tre particularly promising for low-resource settings.

### 3.3. RMR-Tre

Another notable example of trehalose probes is the far-red molecular rotor probe developed by Nicholas Banahene et al. [129].

The RMR-Tre probe is synthesized by conjugating 6-amino-trehalose with a far-red–emitting molecular rotor dye that exhibits turn-on fluorescence in sterically constrained environments such as the mycomembrane, enabling no-wash, rapid detection of live mycobacterial cells. Its major advantage is a 100-fold increase in labeling efficiency compared with earlier probes, with bright far-red emission that minimizes background fluorescence. Specificity testing demonstrated that RMR-Tre efficiently labels mycomembrane-containing bacteria, including *M. smegmatis* and *C. glutamicum*, while showing minimal labeling of mycomembrane-deficient species such as *E. coli* and *B. subtilis*.

In pure cultures, RMR-Tre efficiently labels *M. smegmatis* within 10 min without the need for washing steps. Its compatibility with live-cell imaging makes it a valuable tool for studying mycobacterial membranes and infection dynamics. In addition to the ability of the probe to detect *Mtb* rapidly, the potential of RMR-Tre for DST has also been demonstrated.

In experiments comparing a drug-susceptible wild-type strain of *Mtb* with a kanamycin-resistant strain, RMR-Tre labeling was significantly reduced in the susceptible strain following treatment with rifampicin and kanamycin, reflecting bacterial killing or growth inhibition. Conversely, the kanamycin-resistant strain showed diminished labeling only upon rifampicin treatment, remaining fluorescent despite kanamycin exposure. This differential response demonstrated that RMR-Tre can distinguish drug-resistant bacteria from drug-susceptible bacteria via simple fluorescence detection, providing a direct readout of drug susceptibility.

In an experiment evaluating the potential for DST, the signal obtained from *Mtb* was assessed after 18 h. Given the remarkable properties of this probe, conducting an additional experiment to specifically determine the detection time for *Mtb* would be particularly interesting, as it would allow direct evaluation of the probe’s detection efficiency against the TB pathogen.

## 4. Fluorescent Mycobacteriophages

In recent years, many research groups have developed new approaches based on the use of fluorescent bacteriophages for detecting bacterial pathogens [132,133,134,135,136,137]. Reporter phages are another type of bio reporters uniquely suited for the rapid and sensitive detection of bacterial species. Reporter phages are genetically engineered bacteriophages that contain a reporter gene encoding a fluorescent protein or luciferase (Figure 4) [138]. The bacteriophage interacts with a specific host bacterial cell by binding to surface fragments of the bacterial cell wall through the virus’s receptor-binding proteins [139]. Once attached to the cell, the bacteriophage injects its genome into the cytoplasm, where the DNA undergoes two sequential processes: transcription, in which reporter mRNA is synthesized by bacterial RNA polymerase with the participation of endogenous or exogenous promoters, and translation, in which fluorescent proteins are synthesized on bacterial ribosomes. Expression often occurs at an early stage of the infectious cycle without requiring cell lysis or completion of the lytic phase [30]. After proper folding, the proteins become fluorescently active and generate an optical signal, which enables the detection of viable bacteria. Therefore, the resulting fluorescent signal is amplified intracellularly, allowing for detection. The entire process can be completed within hours, which is much faster than traditional culture-based methods.

### 4.1. Φ2GFP10 Reporter Phage

An example of the application of fluorescent mycobacteriophages is the Φ2GFP10 phage, developed by O’Donnell, Jain et al., that allows for the detection of *Mtb* and DST [140,141]. The recombinant Φ2GFP10 mycobacteriophage was engineered to carry a green fluorescent protein (GFP) reporter gene. Upon infection of mycobacterial cells, the phage expresses GFP, allowing detection through fluorescence microscopy, flow cytometry, or fluorescence plate readers.

In the initial study [140], the Φ2GFP10 phage demonstrated the ability to detect mycobacteria and perform DST for rifampin and kanamycin directly from sputum samples, with results available after 12 h of incubation. Testing for isoniazid and ofloxacin resistance in cultured samples was completed within 36 h, a difference that the authors attributed to the secondary effects of these drugs on the transcriptional and translational processes targeted by the fluorophage.

In subsequent research, particular attention was given to validating the Φ2GFP10 phage assay in sputum samples obtained from patients with HIV [141]. The reporter phage showed high sensitivity for detecting *Mtb* at bacterial loads as low as 10^4^ bacilli per milliliter, including in sputum samples that were smear-negative by conventional microscopy. This represents an important clinical subgroup within HIV-endemic populations, where diagnosis is especially challenging. Moreover, the assay successfully detected rifampin resistance in these samples.

Although the assay exhibited lower specificity, likely due to the potential for the fluorophage to infect NTM, it demonstrated higher sensitivity than the GeneXpert MTB/RIF system, particularly in smear-negative and HIV-coinfected populations.

### 4.2. mCherrybomb Reporter Phage

Another approach for *Mtb* detection based on reporter phages was described by Liliana Rondón et al. [142]. This study reports the development of a fluoromycobacteriophage containing the mCherryBomb gene, enabling rapid detection of viable *Mtb* and determination of rifampicin resistance from sputum. The experimental protocol included sputum decontamination followed by 48 h and 96 h bacterial recovery periods prior to phage infection. The results demonstrated that a 96 h recovery period before phage infection was critical for assay sensitivity, enabling effective detection of low bacterial concentrations in sputum corresponding to 1–19 CFUs on solid media culture. Using the phage method, an average of 8 bacilli per 100 microscopic fields were detected after 96 h of recovery and sensitivity for rifampicin resistance detection also increased from 31.25% at 48 h to 100% at 96 h recovery. These results are explained by the fact that preliminary incubation allows mycobacteria in samples with low bacterial loads to reach a better metabolic state, which facilitates more effective phage infection and fluorescent protein expression. In pure cultures of *Mtb*, the minimum time required for fluorescent signal obtaining using the mCherryBomb phage was 4–6 h [143].

As a derivative of the TM4 phage, the mCherryBomb phage displays a broad host range, infecting not only the *Mtb* but also various NTM. This broad host specificity is consistent with the known characteristics of TM4, which can infect multiple mycobacterial species, including *M. kansasii*, *M. szulgai*, *M. fortuitum*, *M. abscessus*, and *M. avium*. Therefore, to discriminate between *Mtb* and NTM in clinical samples, pretreatment with para-nitrobenzoic acid (PNB) was used. Nevertheless, a primary advantage of this approach is its ability to detect viable mycobacteria and determine rifampicin susceptibility within three to five days of sputum collection, which is substantially faster than conventional culture-based diagnostics, which often require several weeks for results.

### 4.3. Φ2DRM Reporter Phage

A study by Jain et al. presented a dual-reporter mycobacteriophage, Φ2DRM, capable of detecting preexisting persister cells of *Mtb* in sputum samples [144]. The authors engineered the dual-reporter mycobacteriophage Φ2DRM, which contains two fluorescent reporter genes: a red fluorescent protein-encoding gene (tdTomato), whose expression is controlled by promoters associated with persistence-related genes, and a constitutively expressed yellow fluorescent protein gene (mVenus). This design allows the simultaneous detection of bacterial viability (green fluorescence) and the expression of persister-associated genes (red fluorescence).

When exponential-phase *Mtb* cultures were treated with isoniazid (INH), the majority of cells were killed; however, approximately 1% survived as INH-tolerant persisters, exhibiting a distinct transcriptional profile and strong red fluorescence. Importantly, this study demonstrated that this persister-primed subpopulation exists prior to antibiotic exposure in both cultured bacteria and human sputum samples and is enriched following INH treatment.

In clinical sputum samples, the minimum time required to detect the fluorescence of *Mtb* after infection with the Φ2DRM phage was approximately 4 h, as observed by time-lapse microscopy. The fluorescence signal gradually increased and reached a plateau approximately 16–18 h after phage addition. This detection time reflects successful delivery and expression of the fluorescent reporter genes by the phage within *Mtb* cells.

## 5. Discussion

The development of rapid bioimaging reporters represents a promising approach for the rapid and specific detection of *Mtb*. This review discusses the main current categories of rapid reporters designed for no-wash rapid detection, outlining their key advantages and limitations.

Enzyme-activated probes demonstrated high sensitivity and the capability to detect metabolically active mycobacteria. The probes described in this review detect mycobacteria within minutes or hours and do not require additional steps. For some probes, a correlation between the fluorescence intensity and the bacterial load was demonstrated, enabling quantitative analysis. Mycobacteria detection using enzyme-activated probes can be implemented using plate reader, which allows high-throughput screening. Target enzymes of these probes are not encoded in human genome, which minimizes the risk of non-specific interactions.

However, one of the limitations for this type of probes is enzyme heterogeneity: expression of target enzymes may differ among various *Mtb* strains due to distinct polymorphisms and mutations, which in turn can lead to false-negative results.

An additional problem associated with the clinical application of these approaches may be insufficient specificity due to cross-reactivity with NTM, as well as other bacterial species. This can manifest as a low positive signal, potentially leading to false-positive results.

Therefore, further studies are required to systematically evaluate the probe’s cross-reactivity with other bacterial species under varying incubation times and mycobacteria concentration. This investigation should include testing in clinical specimens, primarily sputum, which are routinely used for TB diagnosis.

Another challenge in using probes for TB diagnosis is their natural efflux from bacterial cells, which can shorten the diagnostic window during which the probes remain detectable and thus limit the practical implementation of the approach as a diagnostic test system. One solution to this issue is a strategy that covalently traps the probe inside the cell, as implemented in the CDG-DNB3 probe [81]. CDG-DNB3 achieves this by forming a covalent bond with the DprE1 enzyme, which plays a critical role in lipoarabinomannan and arabinogalactan synthesis. This covalent attachment secures the probe within the bacterial cell, preventing its efflux and thereby preserving signal intensity and detection sensitivity.

Among fluorescent probes, trehalose-based probes have found the widest use in research practice. Their application relies on a mycobacteria-specific trehalose uptake pathway, which transports the probe into the bacterial cell. However, like enzyme-activated probes, trehalose-based probes suffer from cross-reactivity with nontuberculous mycobacteria and related actinomycetes, as well as probe efflux from the cell, imposing limitations on their practical use.

In some studies, pure mycobacterial cultures were used, which does not fully reflect the conditions encountered with clinical specimens. In clinical samples, probe interaction with bacterial cells and their target metabolites may be reduced due to the complex composition of the specimen and suboptimal reaction conditions compared to those achievable with pure cultures.

Reporter phages offer several key advantages, including a rapid turnaround time with results available within hours, high specificity for mycobacteria, and the ability to distinguish metabolically active cells.

An important challenge is selecting a bacteriophage that does not induce premature lysis of mycobacteria, as early cell rupture can reduce fluorescent signal.

Reporter phages’ reliance on active host metabolism to generate signal might be an additional limitation. Dormant or persister forms of *Mtb* may fail to produce a detectable signal, reducing diagnostic sensitivity. Moreover, some phages’ temperature sensitivity and lytic properties can diminish signal and detection sensitivity. The reporter phage approach also faces standardization challenges due to variability in the analytical performance of different phage constructs.

Most of the described reporters interact only with live, metabolically active cells, which is advantageous since these forms of mycobacteria cause active disease. However, in patient specimens non-replicating or dormant mycobacteria may be present in low numbers and remain undetected, leading to false-negative results. To address this limitation, reporter constructs have been engineered to detect both replicating and non-replicating cells. For example, the dual-reporter phage Φ2DRM [144] which contains two fluorescent reporter genes, enabling simultaneous identification of metabolically active and dormant mycobacterial populations.

The performance of conventional microbiological diagnostics—including microscopy, PCR, and culture—depends on bacterial load in the sample. In specific patient populations, such as people living with HIV and children, mycobacteria are often undetectable or present at low concentrations (paucibacillary disease). Among the studies in this review, study of Φ2GFP10 [140,141] evaluated reporter phages in samples from HIV-positive patients and demonstrated sensitivity comparable to the GeneXpert MTB/RIF assay. Moreover, some experiments to date have been conducted exclusively on pure *M. tuberculosis* cultures. Therefore, further validation in clinical specimens, particularly in vulnerable populations such as children and HIV-infected individuals, is needed.

Additionally, the heterogeneity of the methodologies and reported outcomes was observed across the included studies. This variability presents a challenge for direct comparison of time to result and analytical sensitivity. Some studies reported the limit of detection, while others provided only qualitative data—number of cells or colonies detected; time to result in some studies include sample preparation while other studies provide only time of fluorescence detection. Standardized protocols must be established for evaluating fluorescent reporter systems, including determination of limit of detection and time to result. A detailed description of all reagents, consumables, storage conditions, and transport requirements for each assay would enable estimation of per-test costs. Test affordability is a critical determinant of diagnostic access and financial sustainability for national TB programs. Lowering assay costs would expand diagnostic coverage, whereas higher expenses compared to existing methods could pose a major barrier to implementation, particularly in low-income, high-burden settings. Providing all of that information will enable direct comparison of performance and costs both within reporter classes and across different reporter types, representing an important step for advancing these tools into clinical trials.

None of the reporters in this review required bacterial recovery, except for the mCherrybomb reporter phage.

Within the proof-of-principle studies, not only was the feasibility of detecting *Mtb* demonstrated, but preliminary data from individual investigations also showed analytical sensitivity and time to result comparable to microscopy and automated culture-based systems.

The future success of fluorescent reporters will depend on overcoming current limitations through continuous technological innovation, comprehensive clinical validation, and careful consideration of implementation requirements across diverse epidemiological and resource settings. Once optimized and developed into standardized test systems, these probes could potentially be used as dyes in microscopy and serve as a biosensor for the detection of *Mtb* in automated culture-based systems.

## 6. Conclusions

The development of rapid reporters represents an alternative approach to the detection of *Mtb*. Most of the reporters described in this review (Table 1) exhibit advantages such as high sensitivity, rapid real-time detection in clinical samples, and the potential for DST. The elimination of complex sample preparation reduces analysis time and the number of its stages, thereby decreasing the probability of errors and lowering the requirements for personnel qualifications. The simplified protocol reduces hardware requirements, enabling the development of more simple and affordable diagnostic systems based on these approaches.

Although most of the described solutions are still in the proof-of-principle stage, their potential for clinical application has already been demonstrated. To evaluate the possibility of introducing these reporters into clinical practice, it is necessary to assess their analytical and diagnostic characteristics on a larger number of clinical samples. Conducting clinical validation will allow a comprehensive assessment of the effectiveness and feasibility of introducing these approaches into laboratory practice.

## Figures and Tables

**Figure 1 biomedicines-13-02609-f001:**
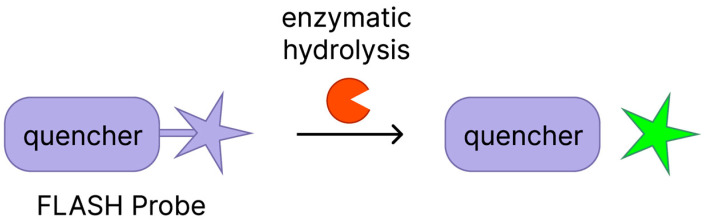
Scheme of the FLASH Hip1 probe activation mechanism. The FLASH Hip1 probe activation mechanism involves enzymatic hydrolysis by Hip1, which cleaves the probe and separates the quencher from the fluorophore. Before hydrolysis, the fluorescence is quenched; after cleavage, fluorescence is restored, generating a detectable signal.

**Figure 2 biomedicines-13-02609-f002:**
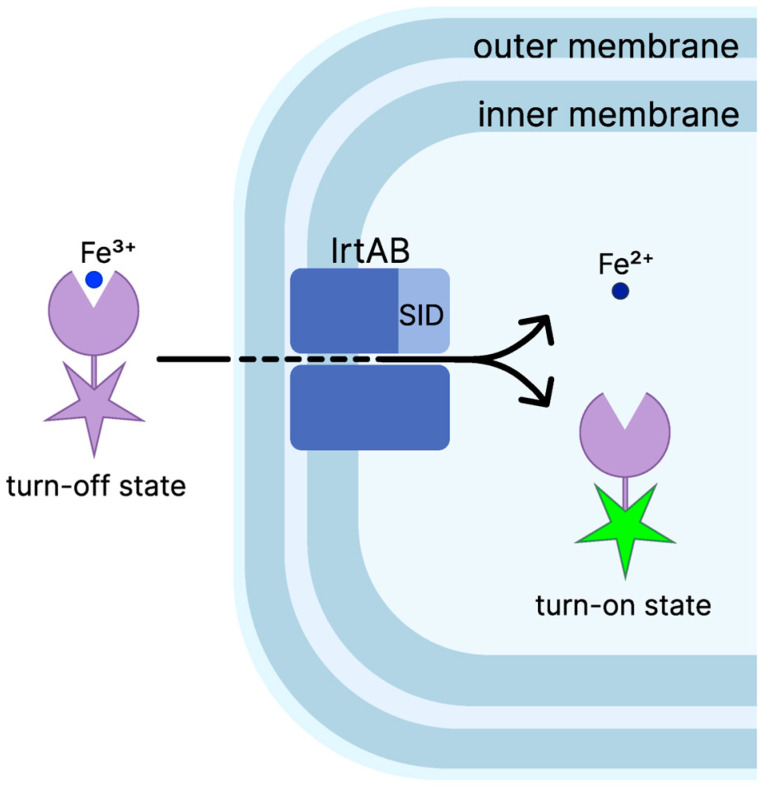
Mechanism of N14G-Fe probe cell envelope penetration with subsequent activation. The N14G-Fe probe enters the cell envelope in its “turn-off” state with Fe^3+^ bound, showing no fluorescence. After transport by the IrtAB complex and passage into the cytosol, Fe^3+^ is reduced to Fe^2+^ and released. This conversion switches the probe to the “turn-on” state, restoring fluorescence. IrtAB: iron-regulated transporter AB; SID: siderophore interaction domain.

**Figure 3 biomedicines-13-02609-f003:**
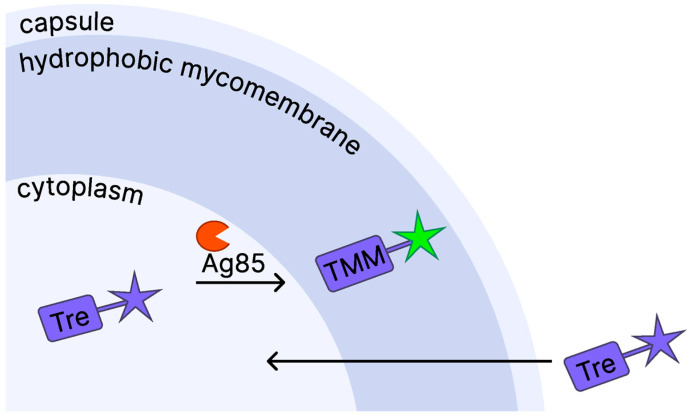
General mechanism of action of trehalose probes for detecting *Mycobacterium tuberculosis.* Trehalose probes are imported into the cytoplasm, where the Ag85 enzyme incorporates the probe into the cell wall, activating the fluorescent signal and enabling visualization of *Mycobacterium tuberculosis.* TMM: trehalose monomycolate; Ag85: antigen 85 complex; Tre: trehalose probe.

**Figure 4 biomedicines-13-02609-f004:**
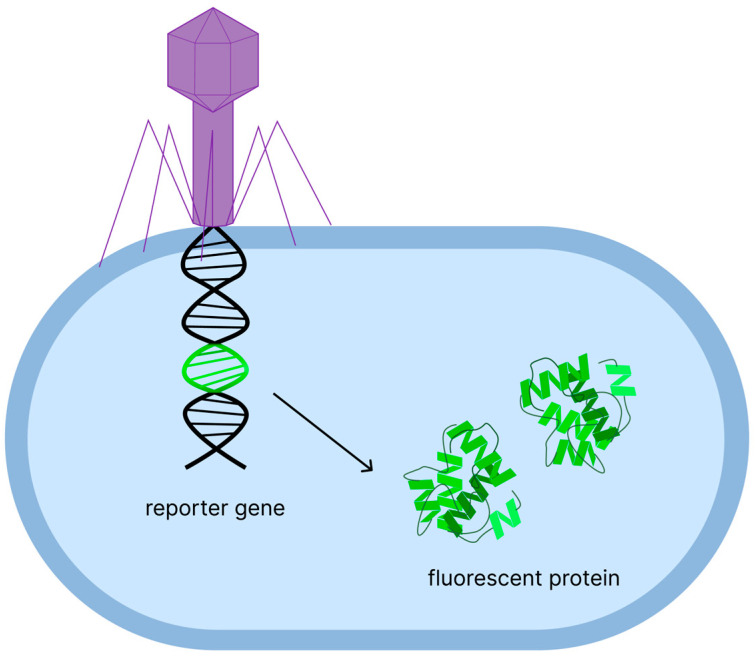
Principle of phage-based pathogen detection. The phage specifically infects target bacteria and injects its genetic material into the bacterial cell, where the reporter gene is expressed, producing fluorescent protein.

**Table 1 biomedicines-13-02609-t001:** “Turn-on” reporter comparison.

Probe/Reporter	Probe Type	Target/Mechanism	Detection Mode	Sample	Time to Result	Detected Pathogen Level	References
FLASH Hip1	Chemiluminescent enzymatic peptide probe	The fluorescence activated by cleavage of the peptide substrate by Hip1 serine protease of *M. tuberculosis*	Chemiluminescence	Processed sputum	1 h	15 × 10^3^ CFU	[80]
CDG-DNB3	Small molecule dual-enzyme-activated fluorogenic probe	Activated by β-lactamase BlaC cleavage; fluorescent product covalently binds DprE1 enzyme	Fluorescence	Processed sputum	1 h	N/D	[81]
NFC Probe (Rv2466c-dependent)	Nitrofuranyl coumarin-based fluorescent probe	Fluorescence activation through elimination of nitro group quenching via enzymatic reduction	Fluorescence	Pure culture	24 h (including incubation)	1.5 × 10^4^ CFU	[96]
Cy3-NO_2_-Tre	Cyanine-based fluorophore (Cy3) linked to trehalose with a nitro group (NO_2_)	Incorporation into mycobacterial cell wall via trehalose uptake pathway; Rv3368c nitroreductase-activated probe	Fluorescence	Processed sputum	15 min	4.3 × 10^2^ CFU	[112]
N14G-Fe	Mycobactin-fluorophore conjugate	Active IrtAB-mediated uptake of iron-chelated probe, intracellular iron reduction releases fluorophore, leading to fluorescence activation	Fluorescence	Processed sputum	10 min	34 CFU	[99]
DMN-Tre	Fluorogenic trehalose analog (solvatochromic)	Incorporation into mycobacterial cell wall via trehalose uptake pathway; fluorescence activation in hydrophobic environment	Fluorescence	Processed sputum	30 min	10^4^ CFU *	[131]
3HC-3-Tre	Fluorogenic trehalose probe (solvatochromic)	Incorporation into mycobacterial cell wall via trehalose uptake pathway; fluorescence activation in hydrophobic environment	Fluorescence	Pure culture	10 min	N/D	[128]
RMR-Tre	Fluorogenic trehalose probe (molecular rotor)	Incorporation into mycobacterial cell wall via trehalose uptake pathway; fluorescence activation in sterically constrained environment	Fluorescence	Pure culture	10 min *	N/D	[129]
Φ2GFP10	Genetically engineered fluorescent reporter phage	Infects of viable *M. tuberculosis* and expresses GFP for visualization	Fluorescence	Processed sputum	12 h ^#^	10^4^ CFU	[140,141]
mCherryBomb	Genetically engineered fluorescent reporter phage	Infects of viable *M. tuberculosis* and expresses red fluorescent protein mCherry	Fluorescence	Processed sputum	16–18 h ^#^	1–19 CFU	[142,143]
Φ2DRM	Genetically engineered fluorescent reporter phage	Infects of viable *M. tuberculosis* and expresses mVenus constitutive, tdTomato for persister cells detection	Fluorescence	Processed sputum	4 h	N/D	[144]

*: for *Mycobacterium smegmatis*. ^#^: Measurements at earlier time points for clinical samples were not evaluated. N/D: not determined for *Mycobacterium tuberculosis.* Hip1: hydrolase important for pathogenesis 1; GFP: green fluorescent protein; DRM: dual-reporter mycobacteriophage; BLC: β-lactamase; DprE1: decaprenylphosphoryl-β-d-ribose-2′-epimerase; IrtAB: iron-regulated transporter AB; CFU: colony-forming units.

## Data Availability

Not applicable.

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
