# Peer review of "Promising Approaches Based on Bioimaging Reporters for Direct Rapid Detection of *Mycobacterium tuberculosis"

_biomedicines, 2025, doi:10.3390/biomedicines13112609_

Round 1
Reviewer 1 Report
Comments and Suggestions for Authors
This review provides a timely and comprehensive overview of promising bioimaging reporter systems for the direct and rapid detection of Mycobacterium tuberculosis (Mtb). The authors successfully synthesize a considerable body of recent research, categorizing the technologies into fluorophore-quencher pairs, fluorogenic trehalose probes, and fluorescent reporter phages. The major strength of the manuscript lies in its systematic organization and its focus on the practical advantages of "no-wash" and "turn-on" mechanisms for future point-of-care diagnostics. However, the review would be significantly strengthened by a more critical analysis of the limitations and challenges facing these technologies and by providing a more unified and consistent comparison of their reported performance metrics to better guide the reader.
Major Issues
-
Lack of a Critical Discussion on Limitations and Challenges: While the review excellently describes the mechanisms and advantages of each technology, it lacks a dedicated, critical section discussing their universal and specific limitations. A deeper analysis is needed on issues such as:
-
Specificity in complex samples: Cross-reactivity with non-tuberculous mycobacteria (NTM) or other microorganisms in real sputum.
-
Probe stability, toxicity, and cost: Key factors for clinical translation and commercial viability.
-
Sensitivity in paucibacillary samples: Performance in samples from HIV-coinfected patients or children, where bacterial load is low.
-
Detection of non-replicating or persister bacteria: The efficacy of these probes against metabolically dormant Mtb populations.
-
-
Insufficient Discussion on the Path towards Clinical Translation: The manuscript focuses heavily on proof-of-concept studies but does not adequately address the pathway to clinical implementation. The conclusion would benefit from a discussion on:
-
The need for standardization and automation of assay protocols.
-
Regulatory hurdles and the requirements for large-scale, multi-center clinical validation.
-
How these technologies would be integrated into or replace existing diagnostic algorithms (e.g., compared to GeneXpert or culture).
-
-
Inconsistent and Non-Comparative Performance Metrics: The data presented in Table 1 and the text are difficult to compare directly. Key parameters are inconsistent:
-
Limit of Detection (LOD): Reported in different units (CFU vs. cells) and for different sample types (pure culture vs. processed sputum). Some entries are marked "N/D" or are for M. smegmatis, limiting the ability to assess true performance for Mtb.
-
Time to Result: The definition varies, sometimes including bacterial recovery/pre-incubation time (e.g., 96 hours for mCherryBomb) and other times only the probe incubation time. This should be clearly defined and standardized for a fair comparison.
-
- Overly Vague Outlook and Future Perspectives: The conclusion states that "clinical validation is needed," but remains generic. The authors should provide a more forward-looking and specific perspective, suggesting which technologies are most promising for specific applications (e.g., point-of-care vs. drug susceptibility testing), and identifying the key technological hurdles that must be overcome for each class of reporter.
Minor Issues
-
Language and Flow: While generally clear, the manuscript would benefit from thorough proofreading to improve sentence flow. Some sentences are long and complex, which can hinder readability. A stylistic revision to enhance conciseness and clarity is recommended.
-
Standardization of Terminology and Abbreviations: Ensure consistency in the use of terms (e.g., use "CFU" consistently) and abbreviations. All abbreviations should be defined upon first use.
- Clarification of "Detection Time": The manuscript should explicitly distinguish between "total assay time" (including sample processing and bacterial recovery) and "probe incubation/reaction time" in both the text and Table 1.
-
Contextualizing Performance: When presenting performance data (e.g., LOD, time), explicitly comparing them to the performance of current gold-standard methods (e.g., "This LOD of X CFU/mL is Y times more sensitive than smear microscopy") would more effectively highlight the advancements.
Author Response
Dear reviewer,
Thank you very much your time considering our manuscript and providing your valuable comments and suggestions!
Please find our answers to your comments:
Major Issues
- Lack of a Critical Discussion on Limitations and Challenges: While the review excellently describes the mechanisms and advantages of each technology, it lacks a dedicated, critical section discussing their universal and specific limitations. A deeper analysis is needed on issues such as:
Thank you for your comment!
We added a ‘discussion’ section to the manuscript (page 11, starting line – 448).
Specificity in complex samples: Cross-reactivity with non-tuberculous mycobacteria (NTM) or other microorganisms in real sputum.
Thank you for the helpful comment!
We agree that cross-reactivity with NTM and other bacteria, including close related actinomycetes, is very important and might result in reduced specificity and false-positive results.
An additional problem associated with the clinical application of these approaches may be insufficient specificity due to cross-reactivity with non-tuberculous mycobacteria, as well as other bacterial species. This can manifest as a low positive signal, potentially leading to false-positive results.
Therefore, further studies are required to systematically evaluate the probe's cross-reactivity with other bacterial species under varying incubation times and mycobacteria concentration. This investigation should include testing in clinical specimens, primarily sputum, which are routinely used for tuberculosis diagnosis.
We added that information to the manuscript (lines 464 – 471)
Probe stability, toxicity, and cost: Key factors for clinical translation and commercial viability.
Thank you for your comment!
Standardized protocols must be established for evaluating fluorescent reporter systems, including determination of limit of detection and time to result. A detailed description of all reagents, consumables, storage conditions, and transport requirements for each assay would enable estimation of per-test costs. Test affordability is a critical determinant of diagnostic access and financial sustainability for national tuberculosis programs. Lowering assay costs would expand diagnostic coverage, whereas higher expenses compared to existing methods could pose a major barrier to implementation, particularly in low-income, high-burden settings. Providing all of that information will enable direct comparison of performance and costs both within reporter classes and across different reporter types, representing an important step for advancing these tools into clinical trials.
We added that information to the manuscript (lines 521 – 530)
Sensitivity in paucibacillary samples: Performance in samples from HIV-coinfected patients or children, where bacterial load is low.
Thank you for your comment!
The performance of conventional microbiological diagnostics—including microscopy, PCR, and culture—depends on bacterial load in the sample. In specific patient populations, such as people living with HIV and children, mycobacteria are often undetectable or present at low concentrations (paucibacillary disease). Among the studies in this review, study of Ф2GFP10 evaluated reporter phages in samples from HIV-positive patients and demonstrated sensitivity comparable to the GeneXpert MTB/RIF assay. Moreover, some experiments to date have been conducted exclusively on pure M. tuberculosis cultures. Therefore, further validation in clinical specimens, particularly in vulnerable populations such as children and HIV-infected individuals, is needed.
We added that information to the manuscript (lines 512 – 520)
Detection of non-replicating or persister bacteria: The efficacy of these probes against metabolically dormant Mtb populations.
Thank you for comment!
Most of the described reporters interact only with live, metabolically active cells, which is advantageous since these forms of mycobacteria cause active disease. However, in patient specimens non-replicating or dormant mycobacteria may be present in low numbers and remain undetected, leading to false-negative results. To address this limitation, reporter constructs have been engineered to detect both replicating and non-replicating cells. For example, the dual-reporter phage Φ2DRM which contains two fluorescent reporter genes, enabling simultaneous identification of metabolically active and dormant mycobacterial populations.
We added that information to the manuscript (lines 504 – 511)
- Insufficient Discussion on the Path towards Clinical Translation: The manuscript focuses heavily on proof-of-concept studies but does not adequately address the pathway to clinical implementation. The conclusion would benefit from a discussion on:
The need for standardization and automation of assay protocols.
Regulatory hurdles and the requirements for large-scale, multi-center clinical validation.
How these technologies would be integrated into or replace existing diagnostic algorithms (e.g., compared to GeneXpert or culture).
Thank you for pointing this out!
Standardized protocols must be established for evaluating fluorescent reporter systems, including determination of limit of detection and time to result. A detailed description of all reagents, consumables, storage conditions, and transport requirements for each assay would enable estimation of per-test costs. Test affordability is a critical determinant of diagnostic access and financial sustainability for national tuberculosis programs. Lowering assay costs would expand diagnostic coverage, whereas higher expenses compared to existing methods could pose a major barrier to implementation, particularly in low-income, high-burden settings. Providing all of that information will enable direct comparison of performance and costs both within reporter classes and across different re-porter types, representing an important step for advancing these tools into clinical trials.
The future success of fluorescent reporters will depend on overcoming current limitations through continuous technological innovation, comprehensive clinical validation, and careful consideration of implementation requirements across diverse epidemiological and resource settings. Once optimized and developed into standardized test systems, these probes could potentially be used as dyes in microscopy and serve as biosensor for detection of M. tuberculosis in automated culture systems.
We added that information to the manuscript (lines 521– 536)
We also replaced the term "proof-of-concept" with "proof-of-principle", which more accurately reflects the current development stage of these reporters.
- Inconsistent and Non-Comparative Performance Metrics: The data presented in Table 1 and the text are difficult to compare directly. Key parameters are inconsistent:
Limit of Detection (LOD): Reported in different units (CFU vs. cells) and for different sample types (pure culture vs. processed sputum). Some entries are marked "N/D" or are for M. smegmatis, limiting the ability to assess true performance for Mtb.
Thank you for your comment, we fully agree with it!
Unfortunately, different units were reported by the authors of various studies, reflecting diverse methodological approaches. We have addressed this issue in the Discussion section (lines 522-528). Due to biosafety considerations and the close genetic relationship between M. smegmatis and M. tuberculosis, many researchers use M. smegmatis as a model organism for specific experimental procedures. Also, its rapid growth enables faster acquisition of results compared to M. tuberculosis. However, findings obtained from M. smegmatis are not always directly applicable to M. tuberculosis due to notable biological and pathogenic differences. In this review, we have prioritized data for M. tuberculosis, but if M. smegmatis was used instead, this is explicitly indicated, ensuring that readers can accurately interpret study-specific results and methodological contexts.
Time to Result: The definition varies, sometimes including bacterial recovery/pre-incubation time (e.g., 96 hours for mCherryBomb) and other times only the probe incubation time. This should be clearly defined and standardized for a fair comparison.
We acknowledge the reviewer's observation regarding the heterogeneity of the methodologies and reported outcomes across the included studies. This variability indeed presents a challenge for direct comparison, as some studies reported the limit of detection while others provided only qualitative data (number of cells or colonies detected). In response to this point, we have now stated this limitation in the revised manuscript. (Lines 520 – 524)
In regards to 96-hours recovery was only needed for mCherryBomb with information provided in lines 410 – 413. For other studies recovery was not used and instead only sputum processing was required.
Overly Vague Outlook and Future Perspectives: The conclusion states that "clinical validation is needed," but remains generic. The authors should provide a more forward-looking and specific perspective, suggesting which technologies are most promising for specific applications (e.g., point-of-care vs. drug susceptibility testing), and identifying the key technological hurdles that must be overcome for each class of reporter.
Thank you for the comment!
We sought to address this question by introducing a dedicated “Discussion” section. In this part, we delineated the technological hurdles associated with each reporter class: enzyme-activated probes (lines 461–480), trehalose-based reporters (lines 481–491), and reporter phages (lines 492–502). Furthermore, within the same section, we highlighted specific challenges related to the translation of these reporter systems, described in the review, into clinical practice (lines 520–541).
Minor Issues
Language and Flow: While generally clear, the manuscript would benefit from thorough proofreading to improve sentence flow. Some sentences are long and complex, which can hinder readability. A stylistic revision to enhance conciseness and clarity is recommended.
Thank you for the comment!
We corrected the sentence, which may have been overly complex, by splitting it into two parts.
Standardization of Terminology and Abbreviations: Ensure consistency in the use of terms (e.g., use "CFU" consistently) and abbreviations. All abbreviations should be defined upon first use.
Thank you for your comment!
We have standardized the terminology and abbreviations
Clarification of "Detection Time": The manuscript should explicitly distinguish between "total assay time" (including sample processing and bacterial recovery) and "probe incubation/reaction time" in both the text and Table 1.
Thank you for your comment!
All reporters, except for reporter-phage mCherryBomb (line 403), doesn’t require recovery or incubation. We have added a reference to this in the Discussion section (line 533).
Contextualizing Performance: When presenting performance data (e.g., LOD, time), explicitly comparing them to the performance of current gold-standard methods (e.g., "This LOD of X CFU/mL is Y times more sensitive than smear microscopy") would more effectively highlight the advancements.
Thank you for the comment!
We made a reference to this in the text (line 535).
Reviewer 2 Report
Comments and Suggestions for Authors
The manuscript entitled: “Promising approaches based on bioimaging reporters for direct rapid detection of Mycobacterium tuberculosis” deal with the very interesting topic pf Mtb diagnosis using bioimaging reporters.
Proper diagnosis of the specific bacterium is of high importance due to the severe impact of Mtb infection especially in low income countries.
In the introduction the authors refer to the existing methods of diagnosis of Mtb with their advantages and disadvantages, while they highlight the importance of developing new methods with high specificity and sensitivity that are moreover easy to apply in the field.
The authors review the literature regarding different type of molecules that can be used as bioimaging reporters. There are only a couple of reviews published after 2019 and this manuscript includes the most recent references in the field reviewing the existing knowledge.
If the authors address some minor issues, the manuscript can be accepted for publication.
• The authors have to deal with some typing errors, e.g. be consistent with writing in italics species names across the manuscript.
• The authors should write figure legends in a more descriptive way, i.e. to explain in 1-2 sentences the mechanism of action of each probe, so that readers can figure out the mechanism.
• In the conclusion the authors write that: “Although most of the described solutions are still in the proof-of-concept stage”. Do they mean that some of the solutions are already used in clinical application? If this is the case, please mention which are those molecules and add this information in table 1.
Author Response
Dear reviewer,
Thank you very much your time considering our manuscript and providing your valuable comments and suggestions!
Please find our answers to your comments:
The authors have to deal with some typing errors, e.g. be consistent with writing in italics species names across the manuscript.
Thank you for your comment!
We corrected these errors in the text.
The authors should write figure legends in a more descriptive way, i.e. to explain in 1-2 sentences the mechanism of action of each probe, so that readers can figure out the mechanism.
Thank you for pointing this out!
We added the explanation of the mechanism of each reporter below the images.
In the conclusion the authors write that: “Although most of the described solutions are still in the proof-of-concept stage”. Do they mean that some of the solutions are already used in clinical application? If this is the case, please mention which are those molecules and add this information in table 1.
Thank you for the comment!
We replaced the term "proof-of-concept" with "proof-of-principle", which more accurately reflects the current development stage of these reporters.